# Doxycycline has distinct apicoplast-specific mechanisms of antimalarial activity

**Megan Okada, Ping Guo, Shai-anne Nalder, Paul A Sigala\***

Department of Biochemistry, University of Utah School of Medicine, Salt Lake City, United States

**Abstract** Doxycycline (DOX) is a key antimalarial drug thought to kill *Plasmodium* parasites by blocking protein translation in the essential apicoplast organelle. Clinical use is primarily limited to prophylaxis due to delayed second-cycle parasite death at 1–3 μM serum concentrations. DOX concentrations > 5 μM kill parasites with first-cycle activity but are thought to involve off-target mechanisms outside the apicoplast. We report that 10 μM DOX blocks apicoplast biogenesis in the first cycle and is rescued by isopentenyl pyrophosphate, an essential apicoplast product, confirming an apicoplast-specific mechanism. Exogenous iron rescues parasites and apicoplast biogenesis from first- but not second-cycle effects of 10 μM DOX, revealing that first-cycle activity involves a metal-dependent mechanism distinct from the delayed-death mechanism. These results critically expand the paradigm for understanding the fundamental antiparasitic mechanisms of DOX and suggest repurposing DOX as a faster acting antimalarial at higher dosing whose multiple mechanisms would be expected to limit parasite resistance.

## Introduction

Malaria remains a serious global health problem, with hundreds of thousands of annual deaths due to *Plasmodium falciparum* parasites. The absence of a potent, long-lasting vaccine and parasite tolerance to frontline artemisinin combination therapies continue to challenge malaria elimination efforts. Furthermore, there are strong concerns that the current COVID-19 pandemic will disrupt malaria prevention and treatment efforts in Africa and cause a surge in malaria deaths that unravels decades of progress (*Weiss et al., 2020*). Deeper understanding of basic parasite biology and the mechanisms of current drugs will guide their optimal use for malaria prevention and treatment and facilitate development of novel therapies to combat parasite drug resistance.

Tetracycline antibiotics like DOX are thought to kill eukaryotic *P. falciparum* parasites by inhibiting prokaryotic-like 70S ribosomal translation inside the essential apicoplast organelle (*Figure 1*; *Dahl et al., 2006*). Although stable *P. falciparum* resistance to DOX has not been reported, clinical use is largely limited to prophylaxis due to delayed activity against intraerythrocytic infection (*Conrad and Rosenthal, 2019*; *Gaillard et al., 2015*). Parasites treated with 1–3 μM DOX, the drug concentration sustained in human serum with current 100–200 mg dosage (*Newton et al., 2005*), continue to grow for 72–96 hr and only die after the second 48 hr intraerythrocytic growth cycle when they fail to expand into a third cycle (*Dahl et al., 2006*). Slow antiparasitic activity is believed to be a fundamental limitation of DOX and other antibiotics that block apicoplast-maintenance pathways (*Gaillard et al., 2015*; *Dahl and Rosenthal, 2007*). First-cycle anti-*Plasmodium* activity has been reported for DOX and azithromycin concentrations > 3 μM, but such activities have been ascribed to targets outside the apicoplast (*Dahl et al., 2006*; *Yeh and DeRisi, 2011*; *Wilson et al., 2015*). A more incisive understanding of the mechanisms and parameters that govern first versus second-cycle DOX activity can inform and improve clinical use of this valuable antibiotic for

**\*For correspondence:**
p.sigala@biochem.utah.edu

**Competing interests:** The authors declare that no competing interests exist.

antimalarial treatment. We therefore set out to test and unravel the mechanisms and apicoplast specificity of first-cycle DOX activity.

## Results

### First-cycle activity by 10 μM DOX has an apicoplast-specific mechanism

Prior studies have shown that 200 μM isopentenyl pyrophosphate (IPP), an essential apicoplast product, rescues parasites from the delayed-death activity of 1–3 μM DOX, confirming an apicoplast-specific target (*Yeh and DeRisi, 2011*). To provide a baseline for comparison, we first used continuous-growth and 48 hr growth-inhibition assays to confirm that IPP rescued parasites from 1 μM DOX (*Figure 2A*) and that DOX concentrations > 5 μM killed parasites with first-cycle activity (*Figure 2A-C* and *Figure 2—figure supplement 1*) as previously reported (*Dahl et al., 2006*). To test the apicoplast specificity of first-cycle DOX activity, we next asked whether 200 μM IPP could rescue parasites from DOX concentrations > 5 μM. We observed that IPP shifted the 48 hr $EC_{50}$ value of DOX from $5 \pm 1$ to $12 \pm 2$ μM (average ± SD of five independent assays, p=0.001 by two-tailed unpaired t-test) (*Figure 2C* and *Figure 2—figure supplement 1*), suggesting that first-cycle growth defects from 5 to 10 μM DOX reflect an apicoplast-specific mechanism but that DOX concentrations > 10 μM cause off-target defects outside this organelle. We further tested this conclusion using continuous growth assays performed at constant DOX concentrations. We observed that IPP fully or nearly fully rescued parasites from first-cycle growth inhibition by 10 μM but not 20 or 40 μM Dox (*Figure 2A and D* and *Figure 2—figure supplement 1*). On the basis of IPP rescue, we conclude that 10 μM DOX kills *P. falciparum* with first-cycle activity by an apicoplast-specific mechanism.

### 10 μM DOX blocks apicoplast biogenesis in the first cycle

Inhibition of apicoplast biogenesis in the second intraerythrocytic cycle is a hallmark of 1–3 μM DOX-treated *P. falciparum*, resulting in unviable parasite progeny that fail to inherit the organelle (*Dahl et al., 2006*). IPP rescues parasite viability after the second cycle without rescuing apicoplast inheritance, such that third-cycle daughter parasites lack the organelle and accumulate apicoplast-targeted proteins in cytoplasmic vesicles (*Yeh and DeRisi, 2011*). We treated synchronized ring-stage D10 (*Waller et al., 2000*) or NF54 (*Swift et al., 2020*) parasites expressing the acyl carrier protein leader sequence fused to GFP ($ACP_L$-GFP) with 10 μM DOX and assessed apicoplast morphology 30–36 hr later in first-cycle schizonts. In contrast to the second-cycle effects of 1–3 μM DOX, the apicoplast in 10 μM DOX-treated parasites failed to elongate in the first cycle. Rescue by 200 μM IPP produced second-cycle parasite progeny with a dispersed GFP signal indicative of apicoplast loss (*Figure 2E* and *Figure 2—figure supplement 2*). We conclude that 10 μM DOX blocks apicoplast biogenesis in the first cycle.

### First- and second-cycle effects of DOX on the apicoplast are due to distinct mechanisms

What is the molecular mechanism of faster apicoplast-specific activity by 10 μM DOX? We first considered the model that both 1 and 10 μM DOX inhibit apicoplast translation but that 10 μM DOX kills parasites faster due to more stringent translation inhibition at higher drug concentrations. This model predicts that treating parasites simultaneously with multiple distinct apicoplast-translation inhibitors, each added at a delayed death-inducing concentration, will produce additive, accelerated activity that kills parasites in the first cycle. To test this model, we treated synchronized D10 parasites with combinatorial doses of 2 μM DOX, 2 μM clindamycin, and 500 nM azithromycin and monitored growth over three intraerythrocytic cycles. Treatment with each antibiotic alone produced major growth defects at the end of the second cycle, as expected for delayed-death activity at these concentrations (*Dahl and Rosenthal, 2007*). Two- and three-way drug combinations caused growth defects that were indistinguishable from individual treatments and provided no evidence for additive, first-cycle activity (*Figure 3A* and *Figure 3—figure supplement 1*). These results contradict a simple model that 1 and 10 μM DOX act via a common translation-blocking mechanism and suggest that the first-cycle activity of 10 μM DOX is due to a distinct mechanism.

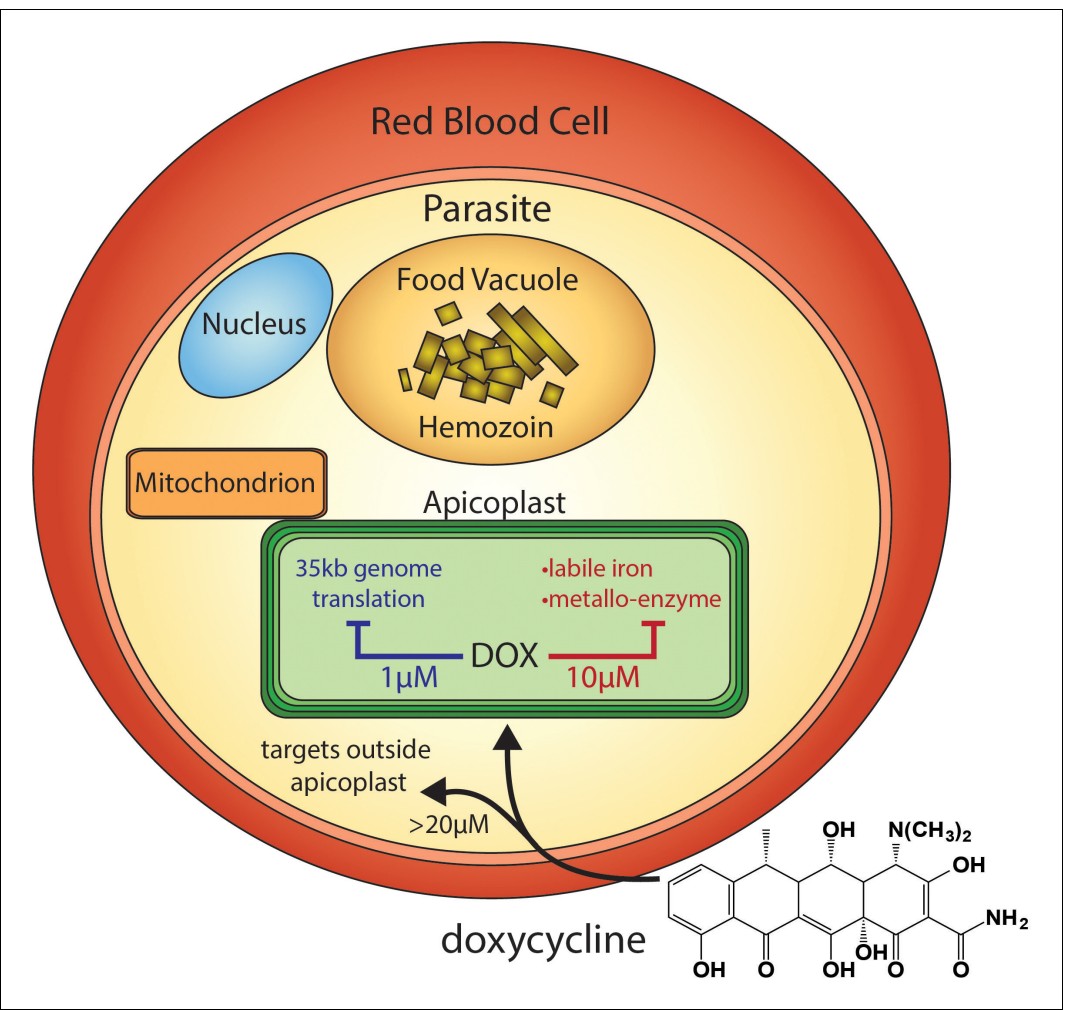

**Figure 1.** Scheme of intraerythrocytic *P. falciparum* parasite depicting doxycycline, its canonical delayed-death mechanism at 1 µM inhibiting apicoplast genome translation, the novel metal-dependent mechanism(s) in the apicoplast explored herein at 10 µM, and off-target activity outside the apicoplast at >20 µM.

## Exogenous iron rescues parasites from first- but not second-cycle effects of 10 µM DOX

Tetracycline antibiotics like DOX tightly chelate a wide variety of di- and trivalent metal ions via their siderophore-like arrangement of exocyclic hydroxyl and carbonyl oxygen atoms (*Figure 1*), with a reported affinity series of $Fe^{3+}>Fe^{2+}>Zn^{2+}>Mg^{2+}>Ca^{2+}$ (*Albert and Rees, 1956*; *Nelson, 1998*). Indeed, tetracycline interactions with $Ca^{2+}$ and $Mg^{2+}$ ions mediate cellular uptake and binding to biomolecular targets such as the tetracycline repressor and 16S rRNA (*Nelson, 1998*; *Orth et al., 2000*). We next considered a model that first-cycle effects of 10 µM DOX reflect a metal-dependent mechanism distinct from ribosomal inhibition causing second-cycle death. To test this model, we investigated whether exogenous metals rescued parasites from 10 µM DOX. We failed to observe growth rescue by 10 µM $ZnCl_2$ (toxicity limit [*Marvin et al., 2012*]) or 500 µM $CaCl_2$ in continuous-growth (*Figure 3B* and *Figure 3—figure supplement 1*) or 48 hr growth-inhibition assays (*Figure 3C*). In contrast, 500 µM $FeCl_3$ (and to a lesser extent 500 µM $MgCl_2$) fully or nearly fully rescued parasites from first-cycle growth inhibition by 10 µM DOX (*Figure 3C and D*), although partial rescue was observed at $FeCl_3$ concentrations as low as 50 µM (*Figure 3—figure supplement 1*). However, parasites treated with 10 µM DOX and 500 µM $FeCl_3$ still succumbed to second-cycle, delayed death (*Figure 3D* and *Figure 3—figure supplement 1*), as expected for distinct mechanisms of first- and second-cycle DOX activity.

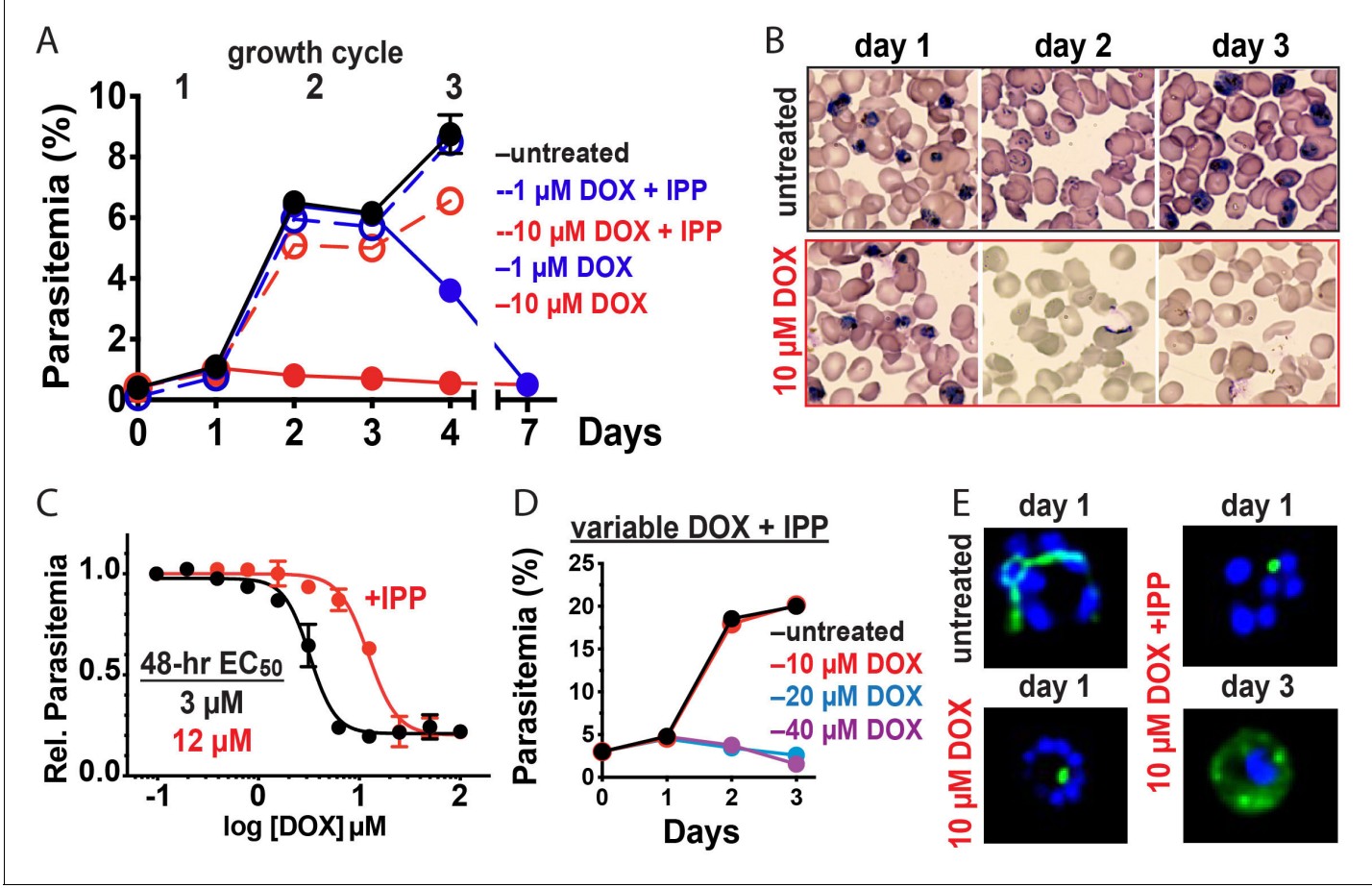

**Figure 2.** 10 μM doxycycline kills *P. falciparum* with first-cycle, apicoplast-specific activity. (A) Continuous growth assay of synchronized Dd2 parasites treated with 1 or 10 μM DOX ±200 μM IPP with (B) Giemsa-stained blood smears for days 1–3. (C) 48 hr growth-inhibition curve for DOX-treated Dd2 parasites ± 200 μM IPP. (D) Continuous growth assay of synchronized Dd2 parasites treated with 10–40 μM DOX and 200 μM IPP. (E) Epifluorescence images of synchronized parasites treated as rings with 10 μM DOX ±200 μM IPP and imaged 36 or 65 hr later (green = $ACP_L$ GFP, blue = nuclear Hoechst stain). Data points in growth assays are the average ± SD of two to four biological replicates. All growth assays were independently repeated two to four times using different batches of blood (shown in *Figure 2—figure supplement 1*).

The online version of this article includes the following figure supplement(s) for figure 2:

**Figure supplement 1.** Additional, independent growth assays with DOX and IPP.

**Figure supplement 2.** Additional epifluorescence images of DOX-treated D10 parasites.

We also observed that 500 μM $FeCl_3$ but not $CaCl_2$ rescued first-cycle apicoplast-branching in 10 μM DOX (*Figure 3E* and *Figure 3—figure supplement 2*). These observations contrast with IPP, which rescued parasite viability in 10 μM DOX but did not restore apicoplast branching (*Figure 2E*). We further noted that $FeCl_3$ selectively rescued parasites from the apicoplast-specific, first-cycle growth effects of 10 μM DOX but did not rescue parasites from the second-cycle effects of 1 μM DOX (*Figure 3F*) or the off-target effects of 20–40 μM DOX (*Figure 3G* and *Figure 3—figure supplement 1*). We conclude that 10 μM DOX kills parasites via a metal-dependent, first-cycle mechanism that blocks apicoplast biogenesis and is distinct from the second-cycle, delayed-death mechanism of 1 μM DOX.

## Discussion

### Metal-dependent mechanisms of first-cycle activity by 10 μM DOX

What is the metal-dependent mechanism of 10 μM DOX, and why is there preferential rescue of parasite growth by $FeCl_3$? Tetracyclines bind iron more tightly than other metals, with an equilibrium

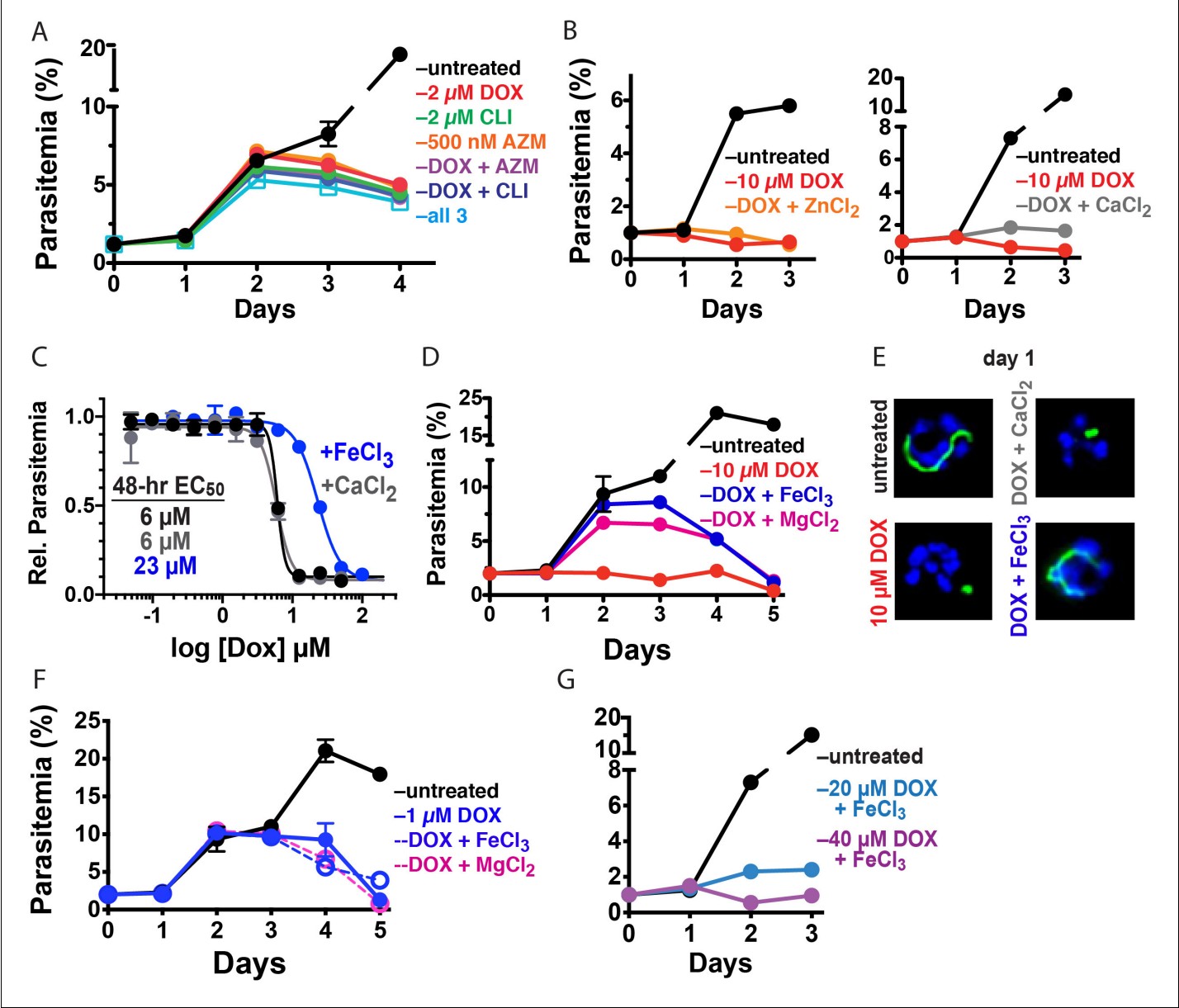

**Figure 3.** 10 µM DOX kills *P. falciparum* with a first-cycle, metal-dependent mechanism. Continuous growth assays of synchronized Dd2 parasites treated with (**A**) DOX, clindamycin (CLI), and/or azithromycin (AZM) and (**B**) 10 µM DOX and 10 µM $ZnCl_2$ or 500 µM $CaCl_2$. (**C**) 48 hr growth inhibition assay of D10 parasites treated with DOX without or with 500 µM $FeCl_3$ or $CaCl_2$. (**D**) Continuous growth assay of synchronized Dd2 parasites treated with 10 µM DOX and 500 µM $FeCl_3$ or $MgCl_2$. (**E**) Epifluorescence images of synchronized parasites treated as rings with 10 µM DOX ±500 µM $FeCl_3$ or $CaCl_2$ and imaged 36 hr later (green = $ACP_L$ GFP, blue = nuclear Hoechst stain). (**F**) Continuous growth assay of synchronized Dd2 parasites treated with 1 µM DOX and 500 µM $FeCl_3$ or $MgCl_2$. (**G**) Continuous-growth assay of synchronized Dd2 parasites treated with 20 or 40 µM DOX and 500 µM $FeCl_3$. Data points in growth assays are the average ± SD of two to four biological replicates. All growth assays were independently repeated using different batches of blood (shown in *Figure 3—figure supplement 1*).

The online version of this article includes the following figure supplement(s) for figure 3:

**Figure supplement 1.** Additional, independent growth assays with DOX, other antibiotics, and metals.

**Figure supplement 2.** Additional epifluorescence images of D10 parasites treated with DOX and metals.

**Figure supplement 3.** Effect of deferoxamine on parasite growth and apicoplast biogenesis.

association constant of $10^{10}$ $M^{-1}$ for 1:1 chelation of $Fe^{3+}$ versus $10^4$ $M^{-1}$ for $Mg^{2+}$ (**Albert and Rees, 1956**). Although the 500 µM concentration of exogenous $FeCl_3$ required for maximal rescue of parasite growth in 10 µM DOX is large relative to the ~1 µM labile iron concentration estimated for the parasite cytoplasm (**Scholl et al., 2005**), the intracellular iron concentration achieved by exogenous addition of 500 µM $FeCl_3$ remains unclear. Indeed, mechanisms of iron uptake and trafficking by blood-stage *P. falciparum* remain sparsely understood (**Scholl et al., 2005**; **Mabeza et al., 1999**), especially uptake across the four membranes that surround the apicoplast.

We first considered whether exogenous $FeCl_3$ might selectively rescue parasites from 10 µM DOX by blocking or reducing its uptake into the parasite apicoplast, since metal chelation has been reported to influence the cellular uptake of tetracycline antibiotics in other organisms (**Nelson, 1998**). However, 500 µM $FeCl_3$ or $MgCl_2$ did not rescue second-cycle parasite death in continuous growth assays with 10 µM (**Figure 3D**) or 1 µM DOX (**Figure 3F**). Furthermore, exogenous iron resulted in only a small, 1.5 µM shift in $EC_{50}$ value from 0.5 to 2 µM in a 96 hr growth inhibition assay, in contrast to the 10.5 µM shift provided by IPP (**Figure 3—figure supplement 1**). These results strongly suggest that DOX uptake into the apicoplast is not substantially perturbed by exogenous iron. The inability of 500 µM $FeCl_3$ to rescue parasites from first-cycle activity by $\geq 20$ µM DOX (**Figure 3G**) further suggests that general uptake of DOX into parasites is not substantially affected by exogenous iron.

We propose two distinct models to explain the metal-dependent effects of 10 µM DOX, both of which could contribute to apicoplast-specific activity. First, DOX could directly bind and sequester labile iron within the apicoplast, reducing its bioavailability for Fe-S cluster biogenesis and other essential iron-dependent processes in this organelle. Indeed, prior work has shown that apicoplast biogenesis requires Fe-S cluster synthesis apart from known essential roles in isoprenoid biosynthesis (**Gisselberg et al., 2013**). In this first model, rescue by exogenous $FeCl_3$ would be due to restoration of iron bioavailability, while modest rescue by 500 µM $MgCl_2$ may reflect competitive displacement of DOX-bound iron to restore iron bioavailability. RPMI growth medium already contains ~400 µM $Mg^{2+}$ prior to supplementation with an addition 500 µM $MgCl_2$, and thus $Mg^{2+}$ availability is unlikely to be directly limited by 10 µM DOX. Consistent with a general mechanism that labile-iron chelation can block apicoplast biogenesis, we observed in preliminary studies that *Plasmodium* growth inhibited by the highly specific iron chelator, deferoxamine (DFO) (**Mabeza et al., 1999**), could be partially rescued by IPP, fully rescued by exogenous $FeCl_3$, and involved a first-cycle defect in apicoplast elongation (**Figure 3—figure supplement 3**). Development of targeted and incisive probes of labile iron within subcellular compartments remains an ongoing challenge in biology (**Breuer et al., 2008**), especially in the *Plasmodium* apicoplast where iron uptake, concentration, and utilization remain sparsely understood. To more broadly evaluate the impact of DOX on iron availability in the apicoplast, we are developing protein-based probes of lipoic acid and isoprenoid biosynthesis, as these two apicoplast-dependent processes require Fe/S-cluster biosynthesis for activity (**Gisselberg et al., 2013**).

In a second model, DOX could bind to additional macromolecular targets within the apicoplast (e.g. a metalloenzyme) via metal-dependent interactions that inhibit essential functions required for organelle biogenesis. Exogenous 500 µM $Fe^{3+}$ would then rescue parasites by disrupting these inhibitory interactions via competitive binding to DOX. This second model would be mechanistically akin to diketo acid inhibitors of HIV integrase like raltegravir that bind to active site $Mg^{2+}$ ions to inhibit integrase activity but are displaced by exogenous metals (**Grobler et al., 2002**; **Hare et al., 2010**). To test this model, we are developing a DOX-affinity reagent to identify apicoplast targets that interact with doxycycline and whose inhibition may contribute to first-cycle DOX activity.

## Conclusions and implications

These results critically expand the paradigm for understanding the fundamental mechanisms of DOX activity against *P. falciparum* malaria parasites. These mechanisms include a delayed, second-cycle defect at 1–3 µM DOX that likely reflects inhibition of 70S apicoplast ribosomes, a first-cycle iron-dependent defect within the apicoplast that uniquely operates at 8–10 µM DOX, and a first-cycle iron-independent mechanism outside the apicoplast at $\geq 20$ µM DOX (**Figure 1**). Pharmacokinetic studies indicate that current 100–200 mg doses of DOX achieve peak human serum concentrations of 6–8 µM over the first six hours which then decrease to 1–2 µM over 24 hr (**Newton et al., 2005**). Although current DOX treatment regimens result in delayed parasite clearance in vivo, both

apicoplast-specific mechanisms of DOX likely operate over this concentration range and contribute to parasite death (*Dahl et al., 2006*). These multiple mechanisms of DOX, together with limited anti-malarial use of DOX in the field, may explain why parasites with stable DOX resistance have not emerged (*Conrad and Rosenthal, 2019*; *Gaillard et al., 2015*).

There has been a prevailing view in the literature that delayed-death activity is a fundamental limitation of antibiotics like DOX that block apicoplast maintenance (*Ramya et al., 2007*; *Kennedy et al., 2019*). Our results emphasize that DOX is not an intrinsically slow-acting antimalarial drug and support the emerging paradigm (*Boucher and Yeh, 2019*; *Amberg-Johnson et al., 2017*; *Uddin et al., 2018*) that inhibition of apicoplast biogenesis can defy the delayed-death phenotype to kill parasites on a faster time-scale. The first-cycle, iron-dependent impacts of 10 µM DOX or 15 µM DFO on apicoplast biogenesis also suggest that this organelle may be especially susceptible to therapeutic strategies that interfere with acquisition and utilization of iron, perhaps due to limited uptake of exogenous iron and/or limited iron storage mechanisms in the apicoplast.

Finally, this work suggests the possibility of repurposing DOX as a faster-acting antiparasitic treatment at higher dosing, whose multiple mechanisms would be expected to limit parasite resistance. Prior studies indicate that 500–600 mg doses in humans achieve sustained serum DOX concentrations $\geq$ 5 µM for 24–48 hr with little or no increase in adverse effects (*Marlin and Cheng, 1979*; *Adadevoh et al., 1976*). DOX is currently contraindicated for long-term prophylaxis in pregnant women and young children, two of the major at-risk populations for malaria, due to concerns about impacts on fetal development and infant tooth discoloration, respectively, based on observed toxicities for other tetracyclines (*Gaillard et al., 2018*). Recent studies suggest that these effects are not associated with short-term DOX use (*Gaillard et al., 2018*; *Todd et al., 2015*; *Cross et al., 2016*), and additional tests can define the safety parameters that would govern short-term use of DOX for treatment in these populations. Recent development of tetracycline derivatives with improved activities may provide another option to deploy this important class of antibiotics for anti-malarial treatment (*Draper et al., 2013*).

# Materials and methods

**Key resources table**

| Reagent type (species) or resource | Designation | Source or reference | Identifiers | Additional information |
|---|---|---|---|---|
| Cell line (*Plasmodium falciparum*) | Dd2 | PMID:1970614 | | |
| Cell line (*Plasmodium falciparum*) | D10 ACP$_L$-GFP | PMID:10775264 | | |
| Cell line (*Plasmodium falciparum*) | NF54-PfMev ACP$_L$-GFP | PMID:32059044 | | |
| Software, algorithm | Prism 8 | GraphPad | RRID:SCR_002798 | |
| Chemical compound, drug | Doxycycline | Sigma-Aldrich | Cat. No. D3447 | |
| Chemical compound, drug | Isopentenyl pyrophosphate | Isoprenoids | Cat. No. IPP001 | |
| Chemical compound, drug | Ferric chloride | Sigma-Aldrich | Cat. No. 236489 | |
| Chemical compound, drug | Clindamycin | Sigma-Aldrich | Cat. No. C6427 | |
| Chemical compound, drug | Azithromycin | Sigma-Aldrich | Cat. No. 75199 | |
| Chemical compound, drug | Deferoxamine | Sigma-Aldrich | Cat. No. D9533 | |

## Materials

All reagents were cell-culture grade and/or of the highest purity available.

## Parasite culture

All experiments were performed using *Plasmodium falciparum* Dd2 (*Wellems et al., 1990*), ACP$_L$-GFP D10 (*Waller et al., 2000*), or ACP$_L$-GFP PfMev NF54 (*Swift et al., 2020*) parasite strains, which were obtained from colleagues and verified by confirming their expected drug sensitivity and/or sequencing strain-specific genetic markers. Parasite culturing was performed as previously described (*Sigala et al., 2015*) in Roswell Park Memorial Institute medium (RPMI-1640, Thermo Fisher 23400021) supplemented with 2.5 g/L Albumax I Lipid-Rich BSA (Thermo Fisher 11020039), 15 mg/L hypoxanthine (Sigma H9636), 110 mg/L sodium pyruvate (Sigma P5280), 1.19 g/L HEPES (Sigma H4034), 2.52 g/L sodium bicarbonate (Sigma S5761), 2 g/L glucose (Sigma G7021), and 10 mg/L gentamicin (Invitrogen Life Technologies 15750060). Cultures were maintained at 2% hematocrit in human erythrocytes obtained from the University of Utah Hospital blood bank, at 37°C, and at 5% $O_2$, 5% $CO_2$, 90% $N_2$. Cultures were mycoplasma-free by PCR test.

## Parasite growth assays

All growth assays were performed with two to four biological replicates (defined according to *Blainey et al., 2014*) in distinct sample wells that were set-up and monitored in parallel. Parasites were synchronized to the ring stage either by treatment with 5% D-sorbitol (Sigma S7900) or by first magnet-purifying schizonts and then incubating them with uninfected erythrocytes for 5 hr followed by treatment with 5% D-sorbitol. Results from growth assays using either of these synchronization methods were indistinguishable within error, and 5% sorbitol was used for synchronization unless stated otherwise.

For continuous growth assays, parasite growth was monitored by diluting sorbitol-synchronized parasites to ~0.5% starting parasitemia, adding additional treatments (antibiotics, IPP, and/or metal salts) at assay initiation, and allowing culture expansion over several days with daily media changes. Growth assays with doxycycline (Sigma D3447), clindamycin (Sigma C6427), and azithromycin (Sigma 75199) were conducted at 0.2% DMSO at the indicated final drug concentration. Growth assays with $ZnCl_2$ (Sigma 208086), $CaCl_2$ (Sigma C4901), $MgCl_2$ (M8266), $FeCl_3$ (Sigma 236489), deferoxamine (Sigma D9533), and/or IPP ($NH_4^+$ salt, Isoprenoids IPP001) were conducted at the indicated final concentrations. Parasitemia was monitored daily by flow cytometry by diluting 10 µl of each parasite culture well from 2 to 3 biological replicates into 200 µl of 1.0 µg/ml acridine orange (Invitrogen Life Technologies A3568) in phosphate buffered saline (PBS) and analysis on a BD FACSCelesta system monitoring SSC-A, FSC-A, PE-A, FITC-A, and PerCP-Cy5-5-A channels.

For $EC_{50}$ determinations via dose-response assay, synchronous ring-stage parasites were diluted to 1% parasitemia and incubated with variable (serially twofold diluted) DOX concentrations ± 200 µM IPP, ±50 µM mevalonate (Cayman 20348), ±500 µM $FeCl_3$, or ±500 µM $CaCl_2$ for 48–120 hr without media changes. Parasitemia was determined by flow cytometry for two to four biological replicates for each untreated or drug-treated condition, normalized to the parasitemia in the absence of drug, plotted as the average ± SD of biological replicates as a function of the log of the drug concentration (in µM), and fit to a four-parameter dose-response model using GraphPad Prism 8.0. All growth assays were independently repeated two to five times on different weeks and in different batches of blood. The 48 hr $EC_{50}$ values determined from five independent assays for DOX ±IPP were averaged and analyzed by unpaired t-test using GraphPad Prism 8.0.

## Fluorescence microscopy

For live-cell experiments, parasites samples were collected at 30–36 or 65 hr after synchronization with magnet purification plus sorbitol treatment (see above). Imaging experiments were independently repeated twice. Parasite nuclei were visualized by incubating samples with 1–2 µg/ml Hoechst 33342 (Thermo Scientific Pierce 62249) for 10–20 min at room temperature. The parasite apicoplast was visualized in D10 (*Waller et al., 2000*) or NF54 mevalonate-bypass (*Swift et al., 2020*) cells using the ACP$_{leader}$-GFP expressed by both lines. Images were taken on DIC/brightfield, DAPI, and GFP channels using either a Zeiss Axio Imager or an EVOS M5000 imaging system. Fiji/ImageJ was used to process and analyze images. All image adjustments, including contrast and brightness, were made on a linear scale. For indicated conditions, apicoplast morphologies in 20–40 total parasites were scored as elongated, focal, or dispersed; counted; and plotted by histogram as the fractional population with the indicated morphology. Analysis of replicate samples indicated standard errors

of the mean that were ≤15% for all samples in the percentage of parasites displaying a given apicoplast morphology in a given condition. Two-tailed unpaired t-test analysis using GraphPad Prism was used to evaluate the significance of observed population differences.

## Acknowledgements
We thank Jeremy Burrows, Dan Goldberg, Don Granger, Daria Hazuda, Jerry Kaplan, Sean Prigge, Dennis Winge and members of the Sigala lab for helpful discussions. PAS holds a Career Award at the Scientific Interface from the Burroughs Wellcome Fund and a Pew Biomedical Scholarship from the Pew Charitable Trusts. Microscopy and flow cytometry were performed using core facilities at the University of Utah.

## Additional information

### Funding

| Funder | Grant reference number | Author |
|---|---|---|
| National Institute of Diabetes and Digestive and Kidney Diseases | T32DK007115 | Megan Okada |
| National Heart and Lung Institute | R25HL108828 | Shai-anne Nalder |
| National Institute of Diabetes and Digestive and Kidney Diseases | U54DK110858 | Paul A Sigala |
| Burroughs Wellcome Fund | 1011969 | Paul A Sigala |
| Pew Charitable Trusts | 32099 | Paul A Sigala |
| National Institute of General Medical Sciences | R35GM133764 | Paul A Sigala |

The funders had no role in study design, data collection and interpretation, or the decision to submit the work for publication.

### Author contributions
Megan Okada, Investigation, Visualization, Writing - review and editing; Ping Guo, Shai-anne Nalder, Investigation; Paul A Sigala, Conceptualization, Supervision, Funding acquisition, Validation, Investigation, Methodology, Writing - original draft, Project administration, Writing - review and editing

### Author ORCIDs
Ping Guo (ID) https://orcid.org/0000-0003-3023-779X
Paul A Sigala (ID) https://orcid.org/0000-0002-3464-3042

### Decision letter and Author response
Decision letter https://doi.org/10.7554/eLife.60246.sa1
Author response https://doi.org/10.7554/eLife.60246.sa2

## Additional files
### Supplementary files
- Source data 1. Source Data for Microscopy Analyses.

- Transparent reporting form

### Data availability
All data reported or described in this manuscript are available and included in the main and supplemental figures and in the microscopy source data file.

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
