## [Decision Letter]

**Acceptance summary:**

This short report provides an example of a potentially better way to deploy a clinically used antimalarial drug. At 1-3 µM serum concentration, doxycycline kills parasites in their second cycle however this delayed parasite clearance is overcome at 10µM. At this higher concentration the parasite are kills in their first cycle by blocking apicoplast biogenesis via a Fe-sensitive target. This important finding could in principle be rapidly incorporated into current practice.

**Decision letter after peer review:**

[Editors’ note: the authors submitted for reconsideration following the decision after peer review. What follows is the decision letter after the first round of review.]

Thank you for submitting your work entitled "Doxycycline has Distinct Apicoplast-Specific Mechanisms of Antimalarial Activity" for consideration by *eLife*. Your article has been reviewed by three peer reviewers, including Jon Clardy as the Reviewing Editor and Reviewer #1, and the evaluation has been overseen by a Senior Editor. The following individuals involved in review of your submission have agreed to reveal their identity: Christopher Goodman (Reviewer #2).

Our decision has been reached after consultation between the reviewers. Based on these discussions and the individual reviews below, we regret to inform you that your work will not be considered further for publication in *eLife*.

The reviewers agreed that the manuscript reported interesting and potentially significant findings, but that in its current state it did not meet the journal's standards. Shortcomings in the mechanistic analysis in what is known to be a complex system, lack of biological repeats, and quantification were primary concerns.

Reviewer #1:

In many fields of medicine improved therapeutic outcomes have resulted more from better ways to deploy clinically used drugs rather than the introduction of brand new drugs. This short report provides another example in the field of malaria: higher doses of doxycycline (DOX), an approved antimalarial drug, avoid its main known drawback, delayed parasite clearance. DOX at 1-3 µM serum concentration dosing – the current standard – kills parasites in their second cycle while 8-10µM dosing kills parasites in their first cycle. The authors report a change in mechanism – emergence of an Fe-sensitive target – as underlying the effect. There are two issues: faster parasite clearance and responsible mechanism. The faster killing data appeared to be quite solid and might have had some precedents in earlier publications, so that part of the report seems solid.

Identifying a mechanism rigorously requires eliminating plausible alternatives, and the authors began by ruling out increased effects at the known ribosomal target. Again, these experiments seemed convincing. DOX, like other tetracycline analogs, bind metals, and the authors add rather large quantities of metals, including Fe at 500 µM, and this result, along with some controls, establishes that a metal-dependent mechanism is likely. It does, at least for me, establish that it's not the same mechanism as lower DOX dose clearance.

Under normal circumstances, I would expect more for an *eLife* paper – a clearer notion of mechanism, something less drastic than a super high metal concentration, and more guidance for future experiments. But these are not normal circumstances, and the potential of this finding to quickly translate into improved therapeutic approaches argues for publication essentially as is.

Reviewer #2:

Summary:

The identification of a fast-acting, apicoplast-related mode of action for doxycycline is significant. It is supported by two independent lines of evidence and helps to explain the absence of doxycycline resistance in the field. The role of metal ions in doxycycline function and bioavailability is complex. Dissecting the effects and proving specific activity requires multiple lines of evidence and robust data. Unfortunately, the data presented does not reach that level. Overall, the quality of the data presented throughout the paper (as outlined in detail below) is not of publishable standard in terms of replication and statistical analysis. The work has real potential but requires more robust data to support the claims.

Essential revisions:

1) The experiments are looking at differences in relatively narrow time frames, but the experimental description provided doesn't give sufficient information about the method of synchronization to assess the experiments. A single sorbitol synchronization produces parasites with an age range from 0 to ~18 hours post invasion, and the distribution of stages can vary between cultures. This large window of parasite age can strongly impact the effects of doxycycline (Dahl et al., 2007). Clarification of this procedure is essential to understanding the experimental results.

2) The absence of replication in the drug trials is not consistent with levels of data expected for publication. The minimal expectation is for three independent biological replicates in each drug trial which allows for statistical comparison. The drug assays presented were only done once (technical replicates…subsection “Parasite Growth Assays:”) which is not sufficient. The small differences seen in effective concentrations require robust statistical analysis. This is highlighted by the 2-fold difference in EC50 for two independent experiments for the Dd2 line (Figure 2—figure supplement 1B and C) which is similar in magnitude to the differences seen with IPP supplementation.

3) The data from the epi-fluorescence imaging requires clarification as to whether all images are from the same experiment and how the synchronization was done. Again, there should be data from multiple replicates, the phenotype needs to be quantified and the results statistically analyzed.

4) The presentation of data for the growth trials is confusing. 2-4 replicates were done but it seems that only one replicate is presented in the text and some others (but not all?) are presented in the supplementary figures. Is it possible to include a single figure with all the replicates and include a statistical analysis?

5) The arguments that the impact of iron chloride is not related to iron sequestering and/or oxidizing Doxycyline, or to a change in function related to altering the metals metals bound to the drug, would be improved by the inclusion of dose-response curves comparing effective concentrations with and without metal supplements. While not definitive they would go a long way to distinguishing between the various possibilities. The single experiment showing little change in growth of parasites under 1µM Doxycycline does not seem enough to address this point.

Reviewer #3:

Doxycycline is an established antimalarial with a fairly well-defined, but complex mechanism of action. The authors carefully studied effects of intermediate concentrations of doxycycline to tease out apicoplast-specific and other effects of the compound. Their results are nicely demonstrated and well-described. The results mostly recapitulate what is already known, but they add the observation that iron rescues parasites from rapid killing by intermediate concentrations of doxycline. This offers what appears to be a valid, but incremental advance in our understanding of the antimalarial action of doxycycline. Some specific concerns are below.

1) Abstract. As mentioned in the text, but not noted in the Abstract, the first cycle activity of doxycycline has already been clearly described (e.g. Dahl et al., 2016; Yeh and DeRis, 2011; Wilson, et al., 2015). Readers of the Abstract will assume incorrectly that the authors are describing a novel result.

2) Abstract. The presentation of a "new paradigm" in the Abstract and Discussion section seems overstated. The MS offers only adds a potential explanation of antimalarial effects of intermediate concentrations of doxycycline; impacts of lower and higher concentrations have already been characterized. Also related to this sentence (and another in the discussion), lack of selection of stable resistance to doxycycline is probably explained by limited use for malaria, as seen with atovaquone/proguanil, another drug that selects readily for resistance in vitro, but has only been used for malaria prophylaxis, and for which resistance is uncommon. The suggestion that the lack of resistance selection is due to a unique mechanism of action is only conjecture, and should probably be omitted from the Abstract.

3) Results section. The sentence is confusing. "…rescues the delayed-death activity…" should be changed to "…rescues parasites from the delayed-death activity…". A similar change is needed in the last paragraph of the Results section.

4) Figure 1 is attractive, but not very helpful for this manuscript, as it shows aspects of plasmodial biology not relevant for this discussion and doesn't clearly inform regarding doxycycline action.

5) Most of Figure 2 represents repeats (albeit nicely performed repeats) and variations of experiments that have been published by other groups.

6) Figure 3. The drug combination experiments are interesting, but interpretation is not as simple as implied. For example, clindamycin appears to lack the high dose rapid killing effects of doxycycline (Dahl and Rosenthal, 2007), so it is difficult to interpret rapid effects of combinations of doxycycline and clindamycin.

7) Discussion section. Mention of repurposing of doxycycline might also refer to studies of analogues, many with far-improved potency, as antimalarials (e.g. PMID: 23629719).

---

## [Author Response]

[Editors’ note: The authors appealed the original decision. What follows is the authors’ response to the first round of review.]

The reviewers agreed that the manuscript reported interesting and potentially significant findings, but that in its current state it did not meet the journal's standards. Shortcomings in the mechanistic analysis in what is known to be a complex system, lack of biological repeats, and quantification were primary concerns.

We thank the reviewers for their careful evaluation and detailed comments. All three reviewers thought our study was interesting and important but raised specific critiques and concerns. We have addressed each of the concerns raised by the reviewers in the revised manuscript and in point-by-point responses to each concern below.

Mechanistic analysis: We have added additional data and analyses to rule out non-specific impacts of metals on DOX uptake or activity, addressed the concern about use of high metal concentrations via additional data and discussion, and provided new data and discussion on unraveling and understanding DOX mechanisms.

Lack of biological repeats: We regret in the prior submission that we mistakenly referred to the parallel samples used in growth assays as “technical replicates” when in fact these replicate samples analyzed in parallel were “biological replicates”. We have clarified in the text and methods that all growth assays data points were based on 2-4 biological replicates (distinct sample wells) but that all assays were also independently repeated on different days and with different blood batches.

Quantification: We have performed additional analyses of the microscopy and growth-assay data to evaluate and establish statistical significance of observed differences.

Reviewer #1:In many fields of medicine improved therapeutic outcomes have resulted more from better ways to deploy clinically used drugs rather than the introduction of brand new drugs. This short report provides another example in the field of malaria: higher doses of doxycycline (DOX), an approved antimalarial drug, avoid its main known drawback, delayed parasite clearance. DOX at 1-3 µM serum concentration dosing – the current standard – kills parasites in their second cycle while 8-10µM dosing kills parasites in their first cycle. The authors report a change in mechanism – emergence of an Fe-sensitive target – as underlying the effect. There are two issues: faster parasite clearance and responsible mechanism. The faster killing data appeared to be quite solid and might have had some precedents in earlier publications, so that part of the report seems solid.Identifying a mechanism rigorously requires eliminating plausible alternatives, and the authors began by ruling out increased effects at the known ribosomal target. Again, these experiments seemed convincing. DOX, like other tetracycline analogs, bind metals, and the authors add rather large quantities of metals, including Fe at 500 µM, and this result, along with some controls, establishes that a metal-dependent mechanism is likely. It does, at least for me, establish that it's not the same mechanism as lower DOX dose clearance.

We thank the reviewer for this feedback and helpful perspective. In our view, the most important finding in our manuscript is that higher-dose DOX can kill parasites quickly (i.e., in the first intraerythrocytic cycle) by an apicoplast-specific mechanism that is distinct from the canonical second-cycle mechanism ascribed to apicoplast ribosome inhibition. The apicoplast-specific effects of DOX have been assumed to be intrinsically slow-acting due to a single mechanism. Our results critically expand this paradigm to clearly establish that higher-dose DOX can specifically block apicoplast biogenesis to kill parasites in the first cycle by a metal-dependent mechanism that is distinct from the canonical mechanism uniquely observed at lower DOX dose. This conclusion of a second, independent apicoplast-specific mechanism, which does not require knowing the specific metal-dependent mechanism, has important therapeutic implications for targeting apicoplast biogenesis and for guiding clinical use of DOX. The importance and timeliness of this conclusion are what drove our decision to submit this work to *eLife* as a short report.

Under normal circumstances, I would expect more for an eLife paper – a clearer notion of mechanism, something less drastic than a super high metal concentration, and more guidance for future experiments. But these are not normal circumstances, and the potential of this finding to quickly translate into improved therapeutic approaches argues for publication essentially as is.

500 µM FeCl_3_ is required for maximal parasite rescue from 10 µM DOX, but we do see partial rescue at iron concentrations as low as 50 µM (Figure 3—figure supplement 1D). While this amount of iron may seem high relative to the 1 µM labile iron estimated in the parasite cytoplasm, it is unclear what effective iron concentration inside the parasite, especially inside the apicoplast, is achieved by exogenous 500 µM iron. Indeed, iron uptake and trafficking mechanisms in bloodstage parasites are very sparsely studied or understood, and exogenous iron would have to traverse 7 membranes (RBC membrane, parasitophorous vacuole membrane, parasite membrane, and 4 apicoplast membranes) to reach the apicoplast matrix.

We agree with the reviewer on the importance and desire to elucidate the specific iron-dependent mechanism of higher-dose DOX in the apicoplast. We have expanded our discussion of possible iron-dependent mechanisms to include preliminary data for studies with the highly-specific iron chelator, deferoxamine (DFO), that support a general mechanism that labile-iron chelation can block apicoplast biogenesis (Figure 3—figure supplement 3). We have on-going studies of DFO and plan to publish a full analysis of these effects elsewhere. We also discuss experimental strategies we are currently using moving forward to test the impact of higher-dose DOX on labile iron availability in the apicoplast as well as to use DOX-affinity reagents to identify apicoplast-specific targets. We have tried to strike a balance on providing some guidance on what approaches we will take to resolve the first-cycle mechanism while not being overly speculative on what the outcomes may be, especially as the core conclusions of this manuscript do not depend critically on knowing the exact metal-dependent mechanism(s).

Although we agree on the importance of understanding mechanism, we also know that demonstration of specific mechanisms can be very challenging and slow. As a point of perspective, we note that apicoplast ribosomal translation inhibition by low-dose DOX has never been formally demonstrated as the specific mechanism of second-cycle parasite death. The observed phenotypes are consistent with this assumed mechanism and the known effects of DOX on bacterial ribosomes but have never been directly shown in *Plasmodium* parasites. Indeed, the Dahl et al., (2006) study, which is rightly regarded as one of the definitive studies of tetracycline effects on the apicoplast, studied apicoplast transcription, not translation.

Reviewer #2:Summary:The identification of a fast-acting, apicoplast-related mode of action for doxycycline is significant. It is supported by two independent lines of evidence and helps to explain the absence of doxycycline resistance in the field. The role of metal ions in doxycycline function and bioavailability is complex. Dissecting the effects and proving specific activity requires multiple lines of evidence and robust data. Unfortunately, the data presented does not reach that level. Overall, the quality of the data presented throughout the paper (as outlined in detail below) is not of publishable standard in terms of replication and statistical analysis. The work has real potential but requires more robust data to support the claims.

We thank the reviewer for this feedback, and we certainly agree on the importance of robust data and analyses. As explained below, we regret in the prior submission that we mistakenly referred to independent, replicate culture wells in growth assays as “technical replicates”, which gave the misimpression that we measured parasitemia values multiple times for a single culture well. In revising the manuscript, we have followed the convention of Blainey et al., (2014) in defining biological replicates as biologically distinct sample wells set-up and monitored in parallel. We have clarified in the text and methods that all growth assays involved 2-4 biological replicate samples that were averaged and plotted ± standard deviation values. Because biological replicates involve growth in the same batch of blood, we have also independently repeated all growth assays on different days and in different batches of blood. We show these independent assay replicates in the supplemental figures to emphasize that our observations were robust across multiple independent experiments.

For microscopy experiments, we have performed the analysis suggested by the reviewer by analyzing 20-40 total parasites in replicate experiments in each condition and using two-tailed unpaired t-tests to analyze and establish the significance of observed population differences. These analyses are now included as supplemental figures.

Essential revisions:1) The experiments are looking at differences in relatively narrow time frames, but the experimental description provided doesn't give sufficient information about the method of synchronization to assess the experiments. A single sorbitol synchronization produces parasites with an age range from 0 to ~18 hours post invasion, and the distribution of stages can vary between cultures. This large window of parasite age can strongly impact the effects of doxycycline (Dahl et al., 2007). Clarification of this procedure is essential to understanding the experimental results.

The Dahl et al., study mentioned by the reviewer used pulsed, 12-hour drug treatments initiated at different delays after culture synchronization. The effects of these short pulses are much more susceptible to variations in culture synchrony.

Our study used continuous drug exposure (i.e., drugs were added immediately after synchronization and maintained continuously) whose effects are expected to be less dependent on culture synchrony. Our initial experiments used a single treatment with 5% D-sorbitol to synchronize parasite, which results in a ~15-hour synchrony window. Culture synchrony and transition between stages were monitored by flow cytometry and blood smear (e.g., Figure 2B). To test the dependence of growth effects from continuous DOX treatment on culture synchrony, we repeated 48- and 96-hour dose-response assays with DOX, IPP, FeCl_3_, and CaCl_2_ using a tighter 5-hour synchrony window obtained by magnet-purifying schizonts, incubating schizonts with fresh, uninfected erythrocytes for 5 hours, and then treating with 5% D-sorbitol (to kill remaining schizonts). The results obtained from synchronization with sorbitol alone versus magnet + sorbitol were indistinguishable within error (see 48-hour EC_50_ curves for Dd2 sychronized by sorbitol alone and D10 synchronized by magnet + sorbitol).

We have revised the text and Materials and methods section to clearly indicate how parasites were synchronized in growth assays and microscopy experiments.

2) The absence of replication in the drug trials is not consistent with levels of data expected for publication. The minimal expectation is for three independent biological replicates in each drug trial which allows for statistical comparison. The drug assays presented were only done once (technical replicates…subsection “Parasite Growth Assays:”) which is not sufficient. The small differences seen in effective concentrations require robust statistical analysis. This is highlighted by the 2-fold difference in EC50 for two independent experiments for the Dd2 line (Figure 2—figure supplement 1B and C) which is similar in magnitude to the differences seen with IPP supplementation.

We regret in the prior submission that we mistakenly referred to independent, replicate culture wells in growth assays as “technical replicates”, which gave the misimpression that we measured parasitemia values multiple times for a single culture well. We have clarified in the text and Materials and methods section that all growth assays involved 2-4 biological replicate samples that were averaged and plotted ± standard deviation values. Because biological replicates involve growth in the same batch of blood, we have repeated all growth assays on different days and in different batches of blood. We show these independent assay replicates in the supplemental figures to emphasize that our observations were robust across multiple independent experiments.

We determined 48-hour EC_50_ values for DOX ±IPP in 4-5 independent experiments (using separate batches of blood and performed on different days). Although the reviewer is correct that the DOX EC_50_ values varied by up to 2-fold between assays, in every individual assay IPP reproducibly shifted the EC_50_ value ≥2-fold to higher value. To evaluate the significance of these differences, we average the EC_50_ values for all assays, calculated a standard deviation, and used an unpaired t-test to evaluate statistical significance. This scatter-plot analysis, pasted below, revealed that 200 µM IPP shifted the DOX EC_50_ value from 5 ±1 µM to 12 ±2 µM, with a P value of 0.001, which strongly supports the significance of this shift. We state this analysis in the Results section and graphically display all values in Figure 2—figure supplement 1C.

3) The data from the epi-fluorescence imaging requires clarification as to whether all images are from the same experiment and how the synchronization was done. Again, there should be data from multiple replicates, the phenotype needs to be quantified and the results statistically analyzed.

The microscopy data is based on 2 independent experiments performed on different days with different blood batches with D10 parasites expressing ACP_L_-GFP and synchronized by magnet purification, reinvasion, and 5% D-sorbitol. We have revised the Materials and methods section to indicate these details. We have performed the analysis suggested by the reviewer by analyzing 20-40 total parasites in replicate experiments in each condition and using two-tailed unpaired t-test to analyze the significance of observed population differences. These analyses are now included as Figure 2—figure supplement 2 and Figure 3—figure supplement 2.

4) The presentation of data for the growth trials is confusing. 2-4 replicates were done but it seems that only one replicate is presented in the text and some others (but not all?) are presented in the supplementary figures. Is it possible to include a single figure with all the replicates and include a statistical analysis?

As explained above, we have clarified in the text and methods that all growth assays involved 2-4 biological replicate samples that were averaged and plotted ± standard deviation values in individual growth assays. Because even biological replicates involve growth in the same batch of blood, we have repeated all growth assays on different days and in different batches of blood. We show these independent assay replicates in the supplemental figures to emphasize that our observations were robust across multiple independent experiments. Because continuous-growth assays performed on different days involved different starting parasitemia values, it is difficult to combine the data into one single representation. Instead, we have included these independent experiments as supplemental figures to demonstrate the reproducibility of our results in different blood batches and in some cases with different parasite strains.

5) The arguments that the impact of iron chloride is not related to iron sequestering and/or oxidizing Doxycyline, or to a change in function related to altering the metals metals bound to the drug, would be improved by the inclusion of dose-response curves comparing effective concentrations with and without metal supplements. While not definitive they would go a long way to distinguishing between the various possibilities. The single experiment showing little change in growth of parasites under 1µM Doxycycline does not seem enough to address this point.

We have performed the experiment suggested by the reviewer and now include 48- and 96-hour dose-response curves for DOX ± 500 µM FeCl_3_ or CaCl_2_ as Figure 3C and Figure 3—figure supplement 1E, along with pertinent discussion of the results. Both experiments fully support the conclusion based on continuous growth assays at 1 or 10 µM DOX that FeCl_3_ but not CaCl_2_ substantially rescues first- but not second-cycle parasite growth in 1 or 10 µM DOX. Iron shifted the 48-hour EC_50_ by 17 µM (which is greater than the 7 µM shift by IPP), but only shifted the 96-hour EC_50_ by 1.5 µM (which is much less than the 10.5 µM shift by IPP).

As discussed in the manuscript, our observation that exogenous iron does not appreciably rescue parasites from the second-cycle effects of 1 or 10 µM DOX strongly suggests that the DOX uptake and availability has not been substantially reduced by iron.

Reviewer #3:Doxycycline is an established antimalarial with a fairly well-defined, but complex mechanism of action. The authors carefully studied effects of intermediate concentrations of doxycycline to tease out apicoplast-specific and other effects of the compound. Their results are nicely demonstrated and well-described. The results mostly recapitulate what is already known, but they add the observation that iron rescues parasites from rapid killing by intermediate concentrations of doxycline. This offers what appears to be a valid, but incremental advance in our understanding of the antimalarial action of doxycycline. Some specific concerns are below.

We thank the reviewer for this feedback. As we note below, the prevailing view in the literature (e.g., Ramya et al., 2007; Kennedy, Crisafulli and Ralph, 2019) is that DOX and other antibiotics that target apicoplast maintenance are fundamentally slow-acting drugs that cause delayed parasite death. The major contribution of our manuscript is our discovery that intermediate concentrations of DOX kill parasites quickly (i.e., in the first cycle) by apicoplast-specific effects that involve a novel iron-dependent mechanism, which has never been shown before. Our study thus critically expands the paradigm (see further explanation below) for understanding the fundamental antimalarial mechanisms of DOX. In our view, this new insight goes well beyond an “incremental advance in our understanding”. We feel strongly that this new understanding is exciting and important, can inform and impact clinical use of DOX and development of new therapeutic strategies that target apicoplast biogenesis, and will be of strong interest to the parasitology, microbial pathogenesis, and cell biology communities that read *eLife*.

1) Abstract. As mentioned in the text, but not noted in the Abstract, the first cycle activity of doxycycline has already been clearly described (e.g. Dahl et al., 2016; Yeh and DeRis, 2011; Wilson, et al., 2015). Readers of the Abstract will assume incorrectly that the authors are describing a novel result.

We have revised the Abstract and manuscript text to clearly indicate that first-cycle activity by DOX was previously known and to focus attention on the novelty of our finding that this faster activity is rescued by exogenous IPP and iron, which has not previously been shown.

2) Abstract. The presentation of a "new paradigm" in the Abstract and Discussion section seems overstated. The manuscript offers only adds a potential explanation of antimalarial effects of intermediate concentrations of doxycycline; impacts of lower and higher concentrations have already been characterized.

We thank the reviewer for this helpful perspective. We have revised the Abstract and Discussion section to state that our results “expand the paradigm for understanding the fundamental antiparasitic mechanisms of DOX.” The prevailing view in the literature (e.g., Ramya et al., 2007; Kennedy, Crisafulli and Ralph, 2019) is that DOX and other antibiotics that target apicoplast maintenance are fundamentally slow-acting drugs that cause delayed parasite death. Our results clarify that apicoplast-specific effects of 10 µM DOX are not intrinsically slow acting and that blocking apicoplast biogenesis can lead to first-cycle parasite death. We agree that the existing paradigm for understanding the action of 1 µM DOX on the apicoplast remains valid. However, our results indicate that this paradigm must be expanded to explain the apicoplast-specific action of 10 µM DOX.

Also related to this sentence (and another in the Discussion section), lack of selection of stable resistance to doxycycline is probably explained by limited use for malaria, as seen with atovaquone/proguanil, another drug that selects readily for resistance in vitro, but has only been used for malaria prophylaxis, and for which resistance is uncommon. The suggestion that the lack of resistance selection is due to a unique mechanism of action is only conjecture, and should probably be omitted from the Abstract.

We agree with the reviewer that limited use of DOX for treatment likely contributes substantially to the lack of parasite DOX resistance. We have removed this statement from the Abstract and have revised the subsection “Conclusions and implications” to state that “multiple mechanisms of DOX, together with limited antimalarial use of DOX in the field, may explain why parasites with stable DOX resistance have not emerged.”

3) Results section. The sentence is confusing. "…rescues the delayed-death activity…" should be changed to "…rescues parasites from the delayed-death activity…". A similar change is needed in the last paragraph of the Results section.

We thank the reviewer for this suggestion, which we agree enhances clarity. We have revised these and other uses in the manuscript to refer to rescue of parasites or parasite growth.

4) Figure 1 is attractive, but not very helpful for this manuscript, as it shows aspects of plasmodial biology not relevant for this discussion and doesn't clearly inform regarding doxycycline action.

We thank the reviewer for this feedback. We have revised Figure 1 to only depict general features of blood-stage *P. falciparum* biology and to graphically summarize the distinct mechanisms of doxycycline elucidated in our study that operate in discrete concentration ranges. Colleagues who reviewed drafts of our manuscript consistently mentioned that a graphical summary of proposed mechanisms was helpful and clarifying. We anticipate that this figure will also be helpful to general readers who may be unfamiliar with blood-stage *P. falciparum*, the apicoplast, and the many membranes that separate the extracellular space from the apicoplast matrix.

5) Most of Figure 2 represents repeats (albeit nicely performed repeats) and variations of experiments that have been published by other groups.

Previous studies have reported second-cycle parasite death by 1 µM DOX, its rescue by IPP, and first-cycle death by DOX concentrations >5 µM. We have clarified in the text that these results were already known. We have repeated these studies in Figure 2 to provide an experimental baseline from which to test the ability of IPP and metals to rescue parasite growth in 10 µM DOX and to show that iron does not rescue second-cycle growth inhibition by 1 or 10 µM DOX. IPP and iron rescue of first-cycle parasite growth in 10 µM DOX has never been shown before our study. We feel strongly that repeating the prior studies is an important control in order to conclude that IPP and iron can selectively rescue parasite growth.

6) Figure 3. The drug combination experiments are interesting, but interpretation is not as simple as implied. For example, clindamycin appears to lack the high dose rapid killing effects of doxycycline (Dahl and Rosenthal, 2007), so it is difficult to interpret rapid effects of combinations of doxycycline and clindamycin.

We would like to clarify that the experimental test in Figure 3A does not require rapid killing by clindamycin in order to test the model that faster parasite death by higher dose DOX is due to more stringent translation inhibition at higher drug combination. These drugs are all proposed to bind to apicoplast ribosomes (CLI and AZM to the 50S subunit, DOX to the 30S subunit) and inhibit protein translation. All of the drug concentrations used in this experiment (2 µM DOX, 2 µM clindamycin, and 500 nM azithromycin) cause second-cycle parasite death when used individually. If delayed, second-cycle death by 1 µM DOX is due to incomplete inhibition of ribosomal translation (and faster action by 10 µM DOX due to more stringent ribosome inhibition), then combination with other ribosome inhibitors (all used at concentrations that cause slow death via presumed ribosome binding) would be predicted to increase the stringency of ribosome inhibition in an additive fashion (akin to increasing DOX concentration alone). Thus, if incomplete ribosome inhibition is the cause of delayed death, then increasing the stringency of inhibition would be predicted to accelerate activity into the first cycle.

Our observation that combination of all three inhibitors at these concentrations does not lead to first-cycle parasite death strongly suggests that increased ribosome inhibition at higher DOX concentration is not the mechanism of faster, first-cycle activity at 10 µM DOX. The additional observations in Figure 3 that iron rescues parasites from first- but not second-cycle death in 1 or 10 µM DOX strongly support our conclusion that first-cycle activity by 10 µM reflects a distinct mechanism from ribosomal inhibition ascribed to 1 µM DOX.

7) Discussion section. Mention of repurposing of doxycycline might also refer to studies of analogues, many with far-improved potency, as antimalarials (e.g. PMID: 23629719).

We thank the reviewer for bringing this study to our attention. We now cite this paper at the end of the subsection “Conclusions and Implications”.